## RESEARCH ARTICLE

# New estimates of the costs of adverse events in patients with cancer

**Achal Patel** [ID]*, **Robert Schuldt** [ID], **Jesse Sussell**

Public Affairs and Access, Genentech, South San Francisco, California, United States of America

* patel.achal@gene.com

## Abstract

The cost of adverse events (AEs) is a critical component of healthcare economic models for cancer treatment; however, few comprehensive estimates of these costs exist. Previous studies focused on costs related to specific AEs, cancer types, or treatment regimens. This study assesses the incremental costs of AEs across 13 cancer types by applying the most current diagnosis schema (International Statistical Classification of Diseases and Related Health Problems [ICD-10]) to patients diagnosed between 2016 and 2022. Data were acquired from US insurance claims and included adult patients who were diagnosed and treated for breast, pancreatic, bladder, lung, prostate, colorectal, skin, liver, head and neck, non-Hodgkin lymphoma, follicular lymphoma, multiple myeloma, or chronic lymphocytic leukemia cancer. We selected 54 AEs for analysis based on their frequency of occurrence, relevance based on severity, and availability of ICD-10 diagnostic codes. We matched patient treatment episodes (the period from initiation of a distinct treatment regimen until discontinuation, change of treatment, or end of data availability) for patients experiencing a specific AE to treatment episodes of similar patients not experiencing that same AE in a 1:1 ratio. We generated estimates for AEs of any severity, severe AEs, and individual tumor types. Our analyses evaluate 392,566 treatment episodes from 110,791 patients. In pooled analyses across tumor types, incremental costs for AEs of any severity (with ≥ 100 treatment episodes) ranged from - $488 for photosensitivity to $29,331 for allergic reaction. Incremental costs for severe AEs (with ≥ 50 treatment episodes) ranged from $16 for dizziness to $18,841 for gastrointestinal bleeding. This study provides a comprehensive assessment of the economic burden of AEs in cancer care and generates a range of estimates to be used as inputs in future economic models.

## 1. Introduction

To improve our understanding of healthcare and health policy, economic models provide insight on the tradeoff between costs and benefits of different therapeutic

**Data availability statement:** This analysis was conducted using commercially licensed medical and pharmacy US insurance claims data. The specific data asset is Pharmetrics Plus, which is licensed by IQVIA. The study team did not receive any special access beyond that granted to other researchers with licensed access to the data. Interested parties may contact IQVIA for access to Pharmetrics Plus: +1 866 267 4479 or https://www.iqvia.com/contact.

**Funding:** This study was funded by Genentech, Inc. The sponsor participated in the design, analysis, and interpretation of the study.

**Competing interests:** All authors have read the Journal's policy and the authors of this manuscript have the following competing interests: Employment and stock or stock options in Genentech, Inc. All support for the present manuscript (e.g., funding, provision of study materials, medical writing, article processing charges, etc.) was provided by Genentech, Inc. This does not alter our adherence to PLOS ONE policies on sharing data and materials.

interventions. In many countries, sophisticated health economic models are required for regulatory approval, along with a demonstration of clinical efficacy. In general, this takes the form of a cost-effectiveness or cost-utility analysis, in which the estimated costs (e.g., cost of additional health screening or preventative medications) and benefits (e.g., overall survival or quality-adjusted life years) of a novel therapeutic approach are weighed against each other. These costs and benefits are usually estimated relative to some comparator, typically the current standard-of-care therapy. For example, in oncology, the costs and benefits of targeted therapy have frequently been modeled relative to chemotherapy or older generations of targeted therapy. In cost-benefit analyses, it is important to measure all costs of a therapeutic approach in addition to the cost of medications [1]. Understanding the tradeoffs between the costs and benefits of different therapies helps payers make decisions under scarcity, thus maximizing the health benefit obtained by the population from a finite payer budget.

In oncology, the primary driver of cost differences is the relative price of the therapeutic regimens. However, adverse events (AEs) represent an important ancillary source of cost differences. For example, an economic study on metastatic melanoma found that the incremental costs associated with specific treatment-related AEs (TRAEs) were high, especially so for AEs related to metabolic and nutritional disorders ($9,135 per 30-day period) [2]. Another direct cost study, related to systemic therapies for melanoma, highlighted large cost differences between different types of grade 3 or 4 TRAEs, especially those that necessitated hospitalizations or expensive outpatient care [3]. Additionally, a cost analysis of immunotherapies in renal cell carcinoma demonstrated a wide spectrum of all-cause and TRAE costs differences between regimens [4].

Conceptually, the expected value cost of an AE is the product of the event rate over time and the average payer cost of treating the event when it occurs. However, there is frequently a paucity of information related to the estimated average cost of treating any individual AE. Parameterization of an economic model with accurate AE cost data can be difficult for a number of reasons: studies that evaluate the costs of AEs frequently only consider AEs experienced by patients with a single tumor type (reducing external validity) and are often conducted using a variety of methodologies (limiting comparability) or use a micro-costing approach, which is known to underestimate total associated costs [5–7]. Frequently, direct or modeled estimates of the incremental cost of treating specific AEs, such as dizziness or sepsis, are often not available. When this occurs, the research community frequently assumes equivalence between the cost of treating plausibly similar AEs (e.g., leukopenia and neutropenia), potentially introducing bias into study results [8–10]. In 2018, Wong et al. significantly improved the data available to the modeling community by publishing a set of estimates of AE costs [5]. Estimates from this study were generated through analysis of insurance claims data and generated data for a pooled sample of patients with cancer, both for patients with overall AEs and for patients with severe (grade 3–4) AEs. That study stratified results for eight different tumor types.

This study fills several gaps currently present in the literature. The timeframe of our dataset (2016−2022) is the most recent among studies considering multiple

tumor types. We have also applied a modern diagnosis schema (International Statistical Classification of Diseases and Related Health Problems [ICD]-10). We generate estimates for more AEs (54 vs 36), stratified across a larger number of distinct tumor types (13 vs 8) relative to the study by Wong et al [5]. Finally, we apply a comprehensive bias reduction strategy (described below) to improve the relative accuracy of our estimates. The results of this study are intended to serve as a resource for the research community engaged in the development of oncology-focused health economic models.

## 2. Materials and methods

### 2.1 Data source

This study was conducted using the IQVIA PharMetrics® Plus closed health plan claims dataset, which includes medical and pharmacy claims for largely commercially insured patients and their dependents, as well as patients with Managed Medicare and Medicare Advantage insurance in the United States.

### 2.2 Overview of study design

In this study, we sought to estimate the incremental cost impact of experiencing different AEs on the total cost of care as measured in healthcare claims data. We considered three different levels of analysis. Our primary analysis was to estimate the cost of individual AEs (e.g., nausea) on healthcare costs in patients with cancer, pooled across different tumor types. In a secondary analysis and consistent with Wong et al. [5], we considered the specific costs of severe AEs (grades 3–4) in the same pooled sample of patients with cancer. In a tertiary analysis, we considered the costs of AEs in 13 groups, stratified by tumor type: breast (ICD-10-CM C50.91), pancreatic (ICD-10-CM C25.1), bladder (ICD-10-CM C67.9), lung (ICD-10-CM C34.9), prostate (ICD-10-CM C61.0), colorectal (ICD-10-CM C18.9), skin (ICD-10-CM C44.9), liver (ICD-10-CM C22.0), head and neck (ICD-10-CM C76.0), non-Hodgkin lymphoma (ICD-10-CM C85.9), follicular lymphoma (ICD-10-CM C82.9), multiple myeloma (ICD-10-CM C90.0), and chronic lymphocytic leukemia (ICD-10-CM C91.1). In each of these analyses, we estimated the incremental cost impact of the AE in question by comparing the total healthcare costs incurred by a "treatment group patient" (i.e., a patient experiencing a given AE type) with those of a "matched control group patient" within a given treatment episode. More details on the matching procedures and treatment episode definitions are described below. Following Wong et al and to reduce potential bias, we excluded all costs associated with antineoplastic treatment, including surgery, radiotherapy, and pharmaceutical costs for oncologic therapies. Although we include treatment regimen in the matching procedure (described below), this exclusion is necessary because there is still potential for a difference in treatment costs due to payer differences, patient geography, or other factors. Since treatment costs are unrelated to the quantity of interest, the incremental cost of AE occurrences, it is simplest to remove them.

### 2.3 Patient selection

**2.3.1 Treatment group selection.** Data were de-identified according to Health Insurance Portability and Accountability Act patient confidentiality requirements, and institutional review board approval was not required. Adult patients were selected for inclusion in the treatment group if they were initially diagnosed with one of the 13 cancer types included in this study between 2016 and 2022 and if they were subsequently diagnosed with one of the 54 AEs of interest within a given treatment episode (see below for treatment episode definitions) (Fig 1).

**2.3.2 Control group selection.** The inclusion criteria for the matching control group were the same as the treatment group, except patients in the control group were required to *not* have a within-treatment episode diagnosis for the AE of interest. For example, for a patient in the treatment group who had the AE *nausea,* the pool of potential matched controls was required to not have nausea but could have any of the other AEs considered in this study.

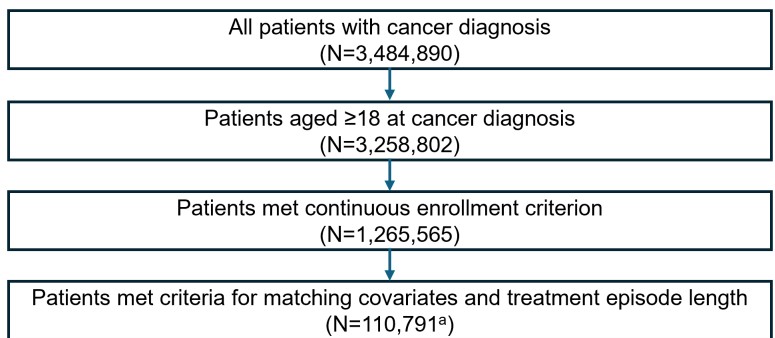

**Fig 1. Attrition table.** ᵃThe 110,791 patients contributed 392,566 treatment episodes.

## 2.4 Outcomes

The study outcome is the average incremental difference in total medical spending between the matched pairs of patients, each of which is defined by either experiencing (treatment group) or not experiencing (matched controls group) the AE of interest during the treatment episode being considered.

## 2.5 Treatment episode definitions

Following Wong et al., each patient was allowed to complete multiple treatment episodes for the analysis [5]. The start of a treatment episode was defined as the initiation of a distinct antineoplastic regimen. The end of a treatment episode was marked by the occurrence of any of the following: discontinuation of the treatment episode (i.e., gap of ≥45 days in the use of all agents that were part of the treatment regimen), initiation of a new antineoplastic regimen (resulting in the start of a new treatment episode), administrative censoring of the patient (end of continuous enrollment), or the end of data availability. Death is not well measured in claims data; thus, deaths in this study were implicitly incorporated as events that precipitated administrative censoring and the end of cost measurement.

## 2.6 Classification and matching of treatment episodes

The matching procedure we use is similar to that described in Wong et al [5]. We began by classifying treatment episodes into two groups: treatment episode with AE and treatment episode without AE. Treatment episode with AE was defined as a diagnosis of an AE recorded during the treatment episode. Treatment episode without AE was defined as the absence of a diagnostic code for the particular AE of interest during the duration of the treatment episode or the 12-month period preceding the start date of the treatment episode.

The set of AEs included in this study was selected first by examining clinical practice guidelines for the types of treatments used for the cancers included in this study. From the list of 361 AEs mentioned in the product inserts of these treatments, AEs were selected/combined for this study based on the frequency of mention, their relevance based on severity, and the availability of respective diagnostic codes from ICD-10-CM. A total of 54 AEs associated with these treatments were included in this study. Following the literature [5,11,12], severe AEs were defined as AEs that required an inpatient stay as recorded by a health insurance claim. AEs of any severity were defined as AEs coded with an ICD-10-CM diagnosis and recorded on a health insurance claim.

Treatment episodes for an AE were matched to similar treatment episodes without that same AE in a 1:1 ratio. A risk to the validity of comparing healthcare costs within treatment episodes is that the duration of treatment episodes will vary from patient to patient; as a result, the time at risk for cost accrual will differ between patients. To control for this potential bias, the duration of the treatment episodes for patients in the matched control group was truncated to be equal to the

duration of the treatment episode for patients experiencing the AE of interest ([Fig 2]). When the control group patient's treatment episode duration was shorter (as opposed to longer) than the treatment group patient's treatment episode duration, that control group member was discarded and a new control was selected through matching. Treatment episodes were matched separately for each AE. Treatment episodes could be matched only once for a specific AE, but they could be matched more than once across AEs.

### 2.7 Statistical considerations

In the main analysis (assessment of AE costs in a pooled population of patients with cancer), we applied three complementary bias reduction techniques. First, patients were exact matched by tumor type. For a patient with tumor type *i*, it was required that their matched control also be diagnosed with tumor type *i*. Patients with multiple cancer diagnoses (e.g., breast and lung) were permitted but matching used the primary diagnosis. Second, for any patient in the treatment group, selection of the paired patient in the matched control group occurred via propensity score matching. Nearest-neighbor matching was used to match treatment and control in a 1:1 ratio with a caliper of 0.2. The following factors were used to determine the propensity score: age, sex, US region, payer type (insurance), product type (insurance), number of patient episodes, line of therapy, index year line of therapy, regimen, metastatic diagnosis pre-Index, adjuvant status pre-index (surgical), stem cell transplant during treatment episode (indicator), and radiotherapy during treatment episode (indicator). Third, following matching, the incremental difference in healthcare costs was estimated in an outcome model, specified

**Treatment episode with a given AE**

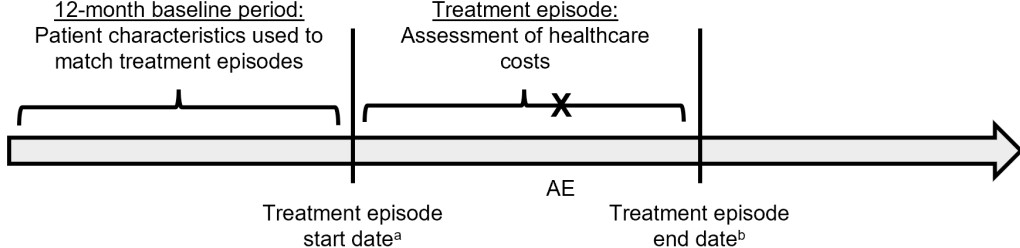

**Matched treatment episode without the given AE**

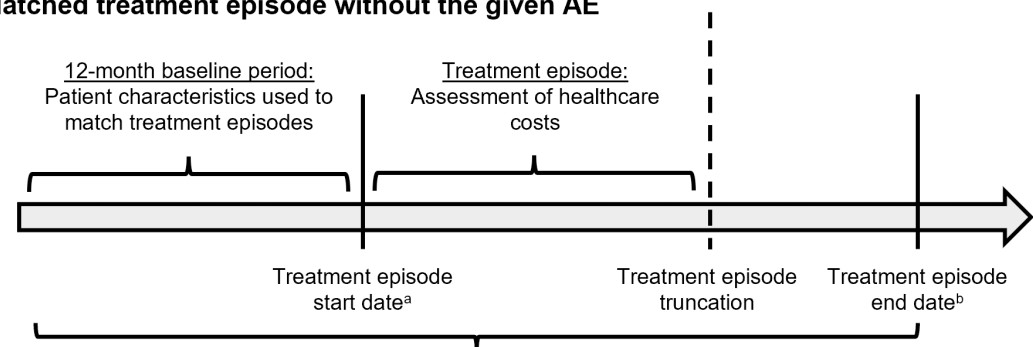

**Fig 2. Adverse event (AE) occurrence in the context of distinct treatment episodes [5].** [a]Treatment episode start date was defined as the date of the initiation of the first antineoplastic agent as part of a treatment regimen. [b]Treatment episode end date was defined as the first of the following events: discontinuation of the treatment regimen (i.e., a gap of ≥45 days in the use of all agents in the treatment), change in treatment regimen, the end of health plan enrollment, or the end of data availability.

using ordinary least squares. The principal purpose of this outcomes model was to control for the count of co-occurring AEs in both members of the matched pair. For example, in estimating the incremental cost impact of nausea, the outcome model incorporated as a covariate the potential co-occurrence of the 53 other AEs within the treatment episode. This covariate was incorporated using a flexible cubic spline approach because co-occurrence of AEs is plausibly correlated—in some cases strongly so. For example, patients with breast cancer who experienced headache/migraine were more likely to experience vomiting, and vice versa (S1 Fig). We note that both the first bias reduction technique (exact matching on tumor type) and the third (controlling for the potential co-occurrence of AEs through regression modeling) were not implemented in the Wong et al. analysis. In particular, the novel use of flexible cubic splines to model the correlation structure across AEs offers a more robust estimate of the incremental cost of any given AE. In addition to controlling for the count of potential co-occurring AEs, the outcomes model also included the Charlson Comorbidity Index as a covariate because comorbidities represent an additional source of potential bias. The outcome models were conducted using robust standard errors to control for the possibility of heteroskedasticity.

The standard assumptions of propensity matching and regression adjustment apply to this analysis. A potential source of bias will remain when there are factors correlated with both the outcome (cost differences) and the treatment (occurrence vs. non-occurrence of AEs of interest), and when those factors are not included in either the propensity model or the outcomes model.

For the secondary analysis (cost estimates for patients experiencing severe AEs), two covariate factors were included: the count of potential co-occurring severe AEs and the count of potential co-occurring AEs, regardless of severity. For example, severe dyspnea and severe pain were correlated in patients with breast cancer (S2 Fig). Additionally, the analysis of severe AE costs required that the AE of interest was the reason for the inpatient stay, which was implemented in the analysis with the requirement that the diagnosis code of interest occurred in the first or second position.

For the tertiary analysis (within-tumor stratified cost estimation), exact matching based on tumor type is implied by design. To perform the propensity score match in the AE analysis, a minimum N value of 100 treatment episodes was required. For the subsequent outcomes model, only AEs with ≥ 50 matched pairs (≥ 100 treatment episodes) were reported.

Finally, we conducted standard regression diagnostics on the outcomes models, including plotting residuals against fitted values to identify instances of nonlinearity. The volume of distinct regression models conducted within this study (1,404) precludes summary reporting of the results of these diagnostic tests.

## 3. Results

### 3.1 Characteristics across matched treatment episodes

The selected 110,791 patients had a total of 392,566 treatment episodes across 54 AEs (Table 1) during the study period from October 1, 2015, to June 30, 2022. The median age was 60 years and 58.3% were females. The most frequent primary cancer was breast cancer (29.9%), followed by cancer of the lung (14.2%), colorectal (11.5%), skin (10.9%), non-Hodgkin lymphoma (6.8%), head and neck (5.3%), pancreatic (5.1%), prostate (4.7%), liver (3.8%), multiple myeloma (3.6%), bladder (2.6%), chronic lymphocytic leukemia (0.9%), and follicular lymphoma (0.7%).

The number of treatment episodes matched across AEs ranged from 241,487 (pain) to 135 (encephalopathy) (Table 2). The five most common AEs of any severity were pain, headache/migraine, vomiting, psychiatric disorders, and colitis (Table 2).

### 3.2 Healthcare costs

The pooled incremental healthcare costs related to AEs of any severity ranged from -$488 for photosensitivity to $29,331 for allergic reaction and application/administration site reaction (Fig 3). The top five most costly AEs of any severity were

**Table 1. Treatment episodes characteristics.**

| Treatment episodes[a] | 392,566 |
|---|---|
| **Age, median (IQR)** | 60 (53, 64) |
| **Female, n (%)** | 228,790 (58.3) |
| **US Region, n (%)** | |
| East | 69,160 (17.6) |
| Midwest | 108,657 (27.7) |
| South | 159,097 (40.5) |
| West | 55,641 (14.2) |
| Other | 11 (<0.1) |
| **Product type, n (%)** | |
| Consumer directed | 9,099 (2.3) |
| HMO | 72,162 (18.4) |
| POS | 23,463 (6.0) |
| PPO | 281,922 (71.8) |
| Unknown | 5,920 (1.5) |
| **Payer type, n (%)** | |
| Commercial | 225,149 (57.4) |
| Medicare Advantage | 36,525 (9.3) |
| Medicare Supplemental | 17,086 (4.4) |
| Self-pay | 111,409 (28.4) |
| Unknown | 2,397 (0.6) |
| **Treatment type, n (%)** | |
| Metastatic pre-treatment | 220,748 (56.2) |
| Adjuvant therapy pre-treatment | 114,051 (29.1) |
| **No. of episodes, n (%)** | |
| 1 | 172,176 (43.9) |
| 2 | 115,546 (29.4) |
| 3 | 104,844 (26.7) |
| **LOT index year, n (%)** | |
| 2016 | 6,955 (1.8) |
| 2017 | 72,665 (18.5) |
| 2018 | 82,780 (21.1) |
| 2019 | 88,387 (22.5) |
| 2020 | 83,708 (21.3) |
| 2021 | 57,587 (14.7) |
| 2022 | 484 (0.1) |
| **LOT number, n (%)** | |
| 1 | 287,210 (73.2) |
| 2 | 78,246 (19.9) |
| 3 | 27,110 (6.9) |
| **Type of studied cancer, n (%)** | |
| Breast | 117,322 (29.9) |
| Lung | 55,834 (14.2) |
| Colorectal | 45,051 (11.5) |
| Skin | 42,804 (10.9) |
| Non-Hodgkin lymphoma | 26,610 (6.8) |
| Head and neck | 20,947 (5.3) |

*(Continued)*

**Table 1.** (Continued)

| Treatment episodes^a | 392,566 |
|---|---|
| Pancreatic | 20,149 (5.1) |
| Prostate | 18,357 (4.7) |
| Liver | 15,070 (3.8) |
| Multiple myeloma | 13,976 (3.6) |
| Bladder | 10,058 (2.6) |
| CLL | 3,518 (0.9) |
| Follicular lymphoma | 2,870 (0.7) |

CLL, chronic lymphocytic leukemia; HMO, health maintenance program; LOT, line of therapy; POS, point of service; PPO, preferred provider organization.

^aAll percentages were calculated with the number of treatment episodes in the denominator. Treatment episodes occurred in 110,791 patients.

allergic reaction and application/administration site reaction ($29,331), gastrointestinal (GI) perforation ($17,366), ketoacidosis ($17,191), lymphocytopenia ($14,854), and GI bleeding ($13,398) (Fig 3).

The pooled incremental healthcare costs associated with severe AEs ranged from $16 for dizziness to $18,841 for GI bleeding (Fig 4). There was no estimate for severe allergic reaction and severe application/administration site reaction to compare with AEs of any severity. The five most costly severe AEs were GI bleeding ($18,841), pain ($16,896), nausea ($13,923), chest pain ($13,093), and headache/migraine ($11,650) (Fig 4).

For common cancer types that represented greater than or equal to 6% of the selected patient population (breast [29.9%], lung [14.2%], colorectal [11.5%], skin [10.9%], non-Hodgkin lymphoma [6.8%]), GI perforation and generalized edema were commonly the most costly AEs of any severity (Table 3). Among these common cancer types, dehydration and diarrhea were generally the least costly AEs of any severity (Table 3). For rarer cancer types (less than 6% of selected patient population) including head and neck (5.3%), pancreatic (5.1%), prostate (4.7%), liver (3.8%), multiple myeloma (3.6%), bladder (2.6%), CLL (0.9%) and follicular lymphoma (0.7%), cost estimates for AEs of any severity had greater variance and were difficult to classify. (Table 4). For example, headaches/migraines cost approximately $15,000–$30,000 in follicular lymphoma, multiple myeloma, and CLL, but less than $1,300 in head and neck, pancreatic, and prostate cancers (Table 4). This is likely a result of the relatively smaller number of matched pairs for these tumor types, and the resulting wider confidence intervals for estimates of mean cost differences.

## 4. Discussion

Following guidance by the Professional Society for Health Economics and Outcomes Research, it is critical to evaluate the incremental cost differences of all types (e.g. AEs) in addition to treatment costs [1]. However, parameterization of the cost of AEs in economic models is often difficult due to limited data and inconsistent or incomplete methodological approaches [5–7]. This real-world study evaluated the incremental healthcare costs associated with AEs among patients diagnosed with 13 cancer types. Based on previous work by Wong et al., this study builds and expands our understanding of cancer-related costs by evaluating a larger number of AEs (54 vs 36), using an updated design schema (ICD-10 vs ICD-9), and a larger number of cancer types (13 vs 8). [5]. This study also flexibly (non-linearly) controls for the influence of co-occurring AEs on healthcare costs in addition to bias control requirements of exact matching on tumor type and propensity matching on patient/clinical characteristics, mitigating potential bias in a more systematic, robust manner than Wong et al.

It is useful to consider the methodology of this study in the context of the existing literature. The ideal measurement of the cost of the occurrence of a given AE in a patient with cancer would be as follows. First, the average total healthcare costs incurred by a group of patients with that AE would be calculated. Second, average total healthcare costs would be calculated for a group of control patients not experiencing the AE but otherwise identical to the first group in every way.

**Table 2. Prevalence and characteristics of matched treatment episodes by AE.**

| Adverse event | Proportion of episodes with AE of any severity (among all episodes), % | Matched episodes, n |
|---|---|---|
| Acute kidney failure | 16.1 | 63,387 |
| Agranulocytosis (neutropenia/leukopenia) | 18.3 | 71,704 |
| Allergic reaction and application/administration site reaction | 0.4 | 1,392 |
| Anemia | 12.3 | 48,432 |
| Arrhythmia | 14.9 | 58,548 |
| Chest pain/angina | 26.4 | 103,722 |
| CNS hemorrhage | 4.9 | 19,283 |
| Colitis | 28.8 | 113,022 |
| Constipation | 3.6 | 14,125 |
| Cystitis/urinary tract infection | 12.4 | 48,804 |
| Dehydration | 23.0 | 90,246 |
| Diarrhea | 4.9 | 19,367 |
| Dizziness | 7.6 | 29,794 |
| Dyspnea | 23.2 | 91,194 |
| Electrolyte imbalance | 27.0 | 105,870 |
| Elevated liver transaminase levels | 0.8 | 3,307 |
| Encephalopathy | <0.1 | 135 |
| Extravasation | <0.1 | 187 |
| Fatigue/generalized weakness/asthenia | 28.7 | 112,659 |
| Generalized edema | 1.0 | 3,771 |
| GI/anal fistula | 0.4 | 1,534 |
| GI bleeding | 2.3 | 8,970 |
| GI perforation | 0.6 | 2,496 |
| Headache/migraine | 53.8 | 211,192 |
| Hearing loss | 0.1 | 546 |
| Heart failure | 1.7 | 6,486 |
| Hematuria | 13.8 | 54,076 |
| Hyperglycemia | 3.8 | 15,072 |
| Hyperlipidemia | 27.6 | 108,210 |
| Hypertension | 0.2 | 758 |
| Hypotension | 0.3 | 1,330 |
| Hypothyroidism/hyperthyroidism | 1.3 | 5,252 |
| Hypoxemia | 4.7 | 18,466 |
| Ketoacidosis | 0.3 | 1,222 |
| Lymphocytopenia | 0.5 | 1,942 |
| Nausea | 22.3 | 87,653 |
| Pain | 61.5 | 241,487 |
| Pancreatitis | 0.9 | 3,457 |
| Pancytopenia | 7.8 | 30,551 |
| Photosensitivity | 0.1 | 508 |
| Pneumonitis/pneumonia | 10.8 | 42,277 |

*(Continued)*

**Table 2.** (Continued)

| Adverse event | Proportion of episodes with AE of any severity (among all episodes), % | Matched episodes, n |
|---|---|---|
| Polyneuropathy | 6.6 | 25,990 |
| Proteinuria | 5.2 | 20,291 |
| Pruritus/erythema/dermatitis | 13.8 | 54,164 |
| Psychiatric disorders | 36.1 | 141,628 |
| Retinal/corneal/sclera problems | 5.8 | 22,931 |
| Sepsis | 11.7 | 45,821 |
| Stomatitis | 2.8 | 10,849 |
| Thrombocytopenia | 13.1 | 51,453 |
| Thromboembolic event | 23.5 | 92,270 |
| Upper respiratory infections | 9.0 | 35,289 |
| Vomiting | 36.3 | 142,541 |
| Weight gain | 0.8 | 2,961 |
| Weight loss | 7.4 | 28,873 |

AE, adverse event; GI, gastrointestinal; CNS, central nervous system.

Finally, the cost of the AE would be calculated as the difference between the two quantities. Because randomization of patients to experience or not experience AEs is not possible, real-world designs such as ours are the only feasible way to perform such analyses.

While there are many such analyses in the extant literature, most of those focus on AEs costs for patients with a specific tumor type, [13–15] sometimes focusing on those treated with a specific regimen [16–18]. Among these, several papers do not adhere to the basic framework we describe above such as using a control group to estimate costs [19,20]. This is an important omission. Cancer patients who do not experience specific AEs typically continue to experience healthcare costs for other reasons, for example costs of cancer treatment, comorbidity management, and treatment of other AEs. Studies that do not utilize a control group will mistakenly include these costs in the estimates of AE costs, biasing them upwards.

Ours is not the first study of AE costs to recognize the importance of co-occurring AEs; Wong et al. described it as a potential explanation for negative point estimates [5]. However, we believe that our study is the first to explicitly control for it in a regression framework. The effect of co-occurring AEs is potentially large. As supplemental S2 Fig shows, among breast cancer patients the correlation between the severe AE "headache/migraine" and the severe AE "vomiting" is 0.34, and the correlation between "dyspnea" and "pain" is even more pronounced, at 0.73. To the extent that these correlations are not controlled for (as we have attempted to do in our study), and to the extent that they represent distinct pathways for cost accrual, estimates of AE costs that follow may be biased.

The results of this study demonstrated that AE-related healthcare costs were considerable for both common and rare cancer types. Although there were commonalities in higher-cost AEs across common cancer types, the actual cost estimates can have a wide range depending on the cancer type. For example, generalized edema is noted as a costly AE in common cancers, with a range from $9,290 in breast cancer to $22,572 in colorectal cancer. Greater variation of AE costs was seen among rare cancer types. For example, the incremental cost per treatment episode of pancytopenia ranged from –$5,662 in liver cancer to $32,941 in follicular lymphoma. These results could be due in part to the inherently smaller patient numbers available for these cancers, making outlier values more influential on the reported means. Larger sample sizes will be needed to better stratify high- and low-cost AEs for rare cancer types. In general, for economic models

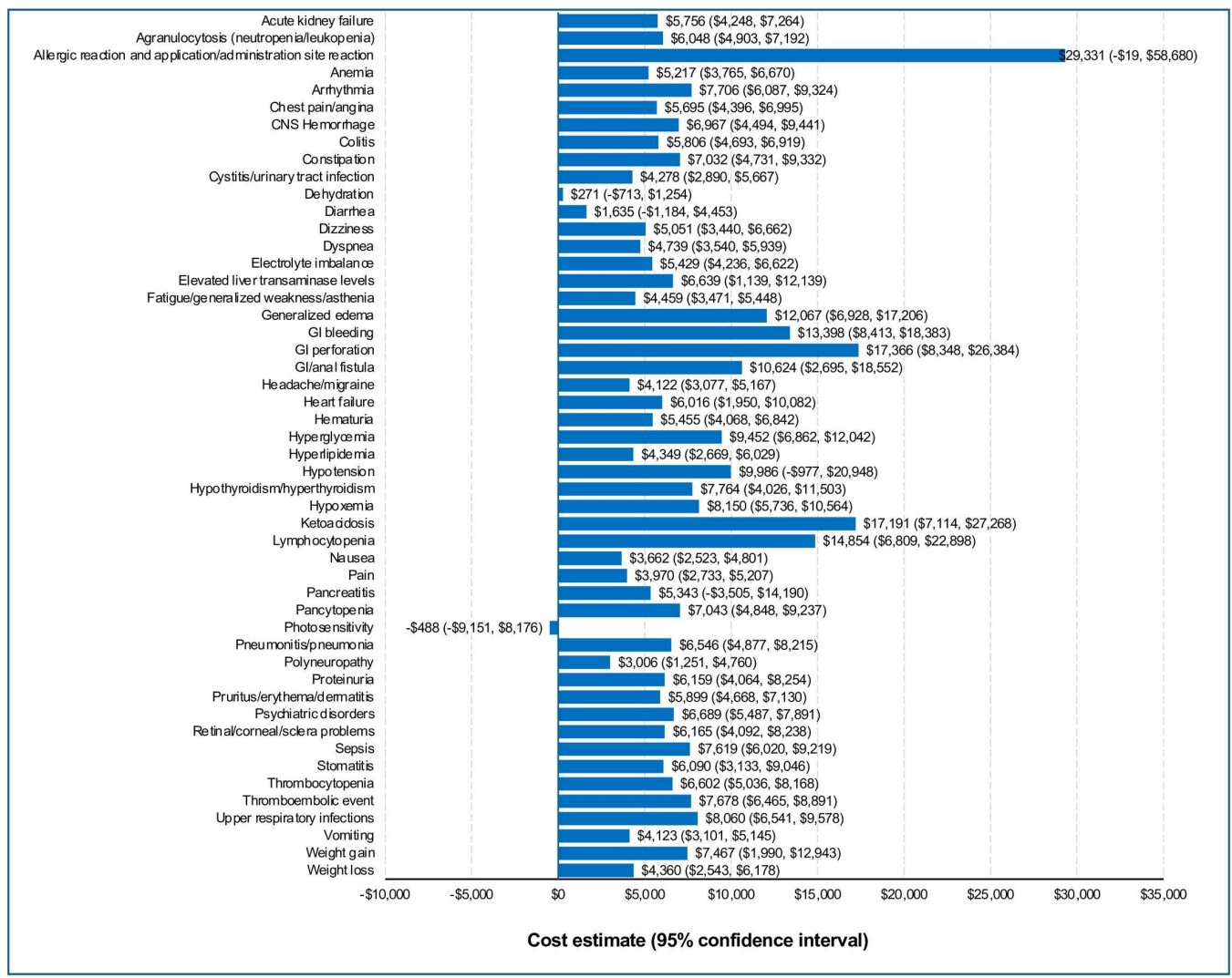

**Fig 3. Incremental costs per matched treatment episode[a]—AEs of any severity, within the pooled sample of patients with cancer.** Incremental costs reported as means (95% confidence interval) in US dollars. AE, adverse event; GI, gastrointestinal; CNS, central nervous system. [a]Only AEs with ≥ 50 matched treatment episode pairs (with ≥ 100 treatment episodes) were included.

involving rare cancers, it may be preferable to use estimates from the pooled models (i.e., estimates pooled across all tumor types, as the benefit of reduced variance may outweigh potential introduced bias).

Finally, estimates from this study that are negative should not be interpreted as evidence that the AE in question is cost-saving, but rather as a reflection of the large, estimated error due to the relatively small number of matched pairs. For example, in general, negative sample estimates are more likely to occur when the true population mean difference is closer to zero. As such, negative point estimates here should be interpreted with caution. Another potential explanation is that, in order to boost sample size, our study allowed AEs to occur in any diagnosis position. Negative point estimates could be partially explained by matching in which control patients who have the AE in question also have another, more expensive condition in a higher diagnosis line. A final possible explanation for negative point estimates is the occurrence of AEs not included in our study design. For example, across all cancers, we estimated the incremental cost of the AE

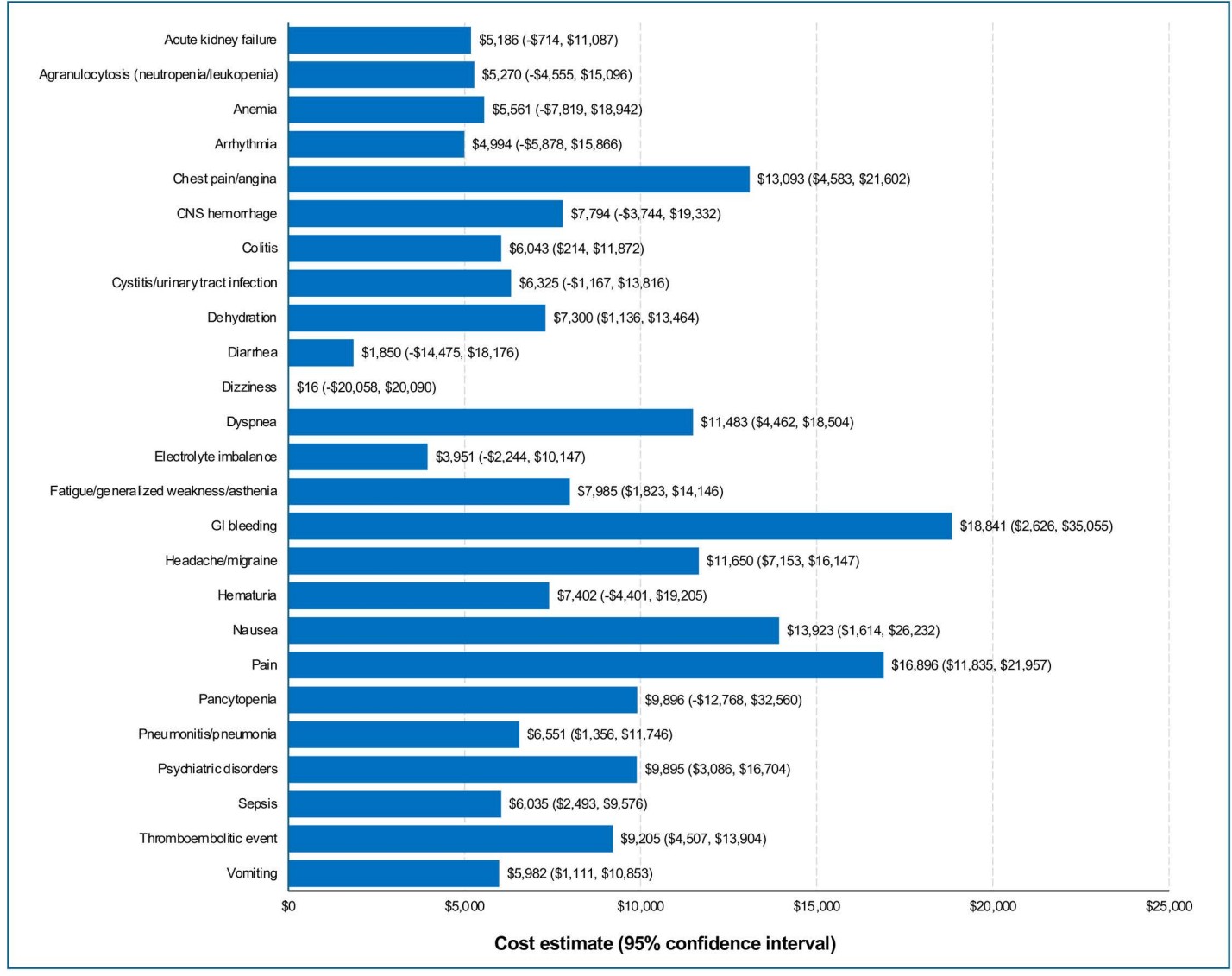

**Fig 4. Incremental costs per matched treatment episode[a]—severe AEs, within the pooled sample of patients with cancer.** Incremental costs reported as means (95% confidence interval) in US dollars. AEs, adverse events; CNS, central nervous system; GI, gastrointestinal. [a]Only severe AEs (grade 3–4) with ≥ 25 matched treatment episode pairs (with ≥ 50 treatment episodes) were included.

photosensitivity to be -$488. If photosensitivity has a strong negative correlation with an AE not included in our design, and if the cost of treating that AE is large, then our estimates of the cost of treating photosensitivity in these patients would be biased downwards. This suggests that the co-occurrence of AEs is an important area for future research.

Severe AEs tend to cost more than less severe AEs due to the associated hospital stay inherent in the definition of a severe AE. This holds true in this analysis where the mean cost of any AE is $6,697 vs $9,305 for a severe AE. However, the costliest AE of any severity was $29,331 for allergic reaction which was higher than the highest-cost severe AE (GI bleeding, $18,841). This difference is in part due to the lack of an estimate for severe allergic reaction, so no direct comparison is possible in this analysis. Additionally, severe AE estimates were generated from a much smaller set of treatment episodes than that for AEs of any severity, leading to greater variability of estimates.

**Table 3. Mean incremental costs (in US dollars) per treatment episode—AEs of any severity, stratified by common tumor type (≥6% of selected patients). Each n indicates the number of treatment episodes.**

| Adverse event, $ | Breast (n = 117,322) | Lung (n = 55,834) | Colorectal (n = 45,051) | Skin (n = 42,804) | Non-Hodgkin lymphoma (n = 26,610) |
|---|---|---|---|---|---|
| Acute kidney failure | 8,895.76 | 4,259.68 | 7,190.18 | 3,505.82 | 12,433.98 |
| Agranulocytosis (neutropenia/leukopenia) | 5,413.75 | 2,065.66 | 10,902.74 | 3,943.75 | 9,962.04 |
| Allergic reaction and application/administration site reaction | 12,903.34 | – | – | 65,814.34 | 26,601.09 |
| Anemia | 5,687.90 | 4,027.30 | 3,132.44 | 6,613.15 | 7,298.97 |
| Arrhythmia | 8,537.93 | 11,627.40 | 5,582.48 | 7,951.48 | 11,222.62 |
| CNS hemorrhage | 10,954.08 | 6,763.96 | 5,402.52 | 7,108.33 | 17,604.06 |
| Chest pain/angina | 10,480.51 | 6,162.70 | −1,637.54 | 6,566.36 | 7,866.06 |
| Colitis | 7,346.59 | 4,351.42 | −1,145.71 | 7,756.32 | 12,742.22 |
| Constipation | 10,345.46 | 7,193.30 | 6,716.07 | 5,784.16 | 4,683.02 |
| Cystitis/urinary tract infection | 5,557.54 | 4,835.92 | −633.38 | 6,106.31 | 10,242.15 |
| Dehydration | 1,574.27 | 1,661.09 | −1,671.00 | 381.44 | 3,684.12 |
| Diarrhea | −1,562.81 | 1,272.80 | −116.61 | 8,675.80 | 2,178.41 |
| Dizziness | 7,754.53 | 5,216.65 | 399.32 | 6,432.51 | 9,490.52 |
| Dyspnea | 6,805.41 | 3,450.07 | 985.94 | 4,033.81 | 5,892.07 |
| Electrolyte imbalance | 7,309.92 | 4,941.51 | 4,458.96 | 5,950.57 | 9,953.76 |
| Elevated liver transaminase levels | 6,408.31 | 4,476.71 | 1,708.77 | 11,086.15 | 20,852.48 |
| Encephalopathy | −11,849.00 | – | – | – | – |
| Extravasation | 415.85 | – | – | – | |
| Fatigue/generalized weakness/asthenia | 6,777.15 | 4,260.47 | 504.77 | 5,846.86 | 7,129.52 |
| GI/anal fistula | 3,311.63 | – | 12,135.65 | 364.85 | – |
| GI bleeding | 14,728.61 | 8,803.66 | 2,796.57 | 1,800.80 | 12,235.31 |
| GI perforation | 18,223.13 | 23,428.75 | 16,836.23 | 12,803.07 | – |
| Generalized edema | 9,290.14 | 12,217.72 | 22,572.37 | 14,982.55 | 11,667.39 |
| Headache/migraine | 6,859.83 | 4,670.86 | −553.62 | 6,661.58 | 4,508.23 |
| Hearing loss | 5,035.74 | – | – | – | – |
| Heart failure | −300.52 | 5,717.86 | −2,199.89 | 5,378.08 | 10,881.46 |
| Hematuria | 8,024.32 | 5,627.13 | 1,794.61 | 4,679.72 | 9,027.99 |
| Hyperglycemia | 9,496.42 | 8,207.57 | 9,444.42 | 10,125.02 | 18,396.79 |
| Hyperlipidemia | 5,033.31 | 9,329.02 | 4,900.21 | 3,195.61 | 5,061.72 |
| Hypertension | 9,191.68 | – | – | – | – |
| Hypotension | 8,312.95 | 10,615.24 | – | – | – |
| Hypothyroidism/hyperthyroidism | 10,124.33 | 5,152.00 | 3,178.06 | 10,073.85 | 7,646.64 |
| Hypoxemia | 11,299.67 | 7,080.79 | 1,253.62 | 10,523.62 | 4,819.06 |
| Ketoacidosis | 12,064.15 | 5,794.78 | – | 22,940.21 | 27,767.77 |
| Lymphocytopenia | 9,698.70 | 16,253.98 | – | 7,485.89 | 16,060.83 |
| Nausea | 5,807.81 | 8.31 | 1,748.84 | 3,861.25 | 4,711.33 |
| Pain | 5,166.13 | 8,324.04 | 3,027.68 | 3,680.63 | 5,203.78 |
| Pancreatitis | 12,436.07 | 12,205.91 | 3,941.58 | 11,718.00 | 23,656.66 |
| Pancytopenia | 6,684.39 | 2,968.81 | 6,398.27 | 1,544.49 | 14,010.04 |
| Photosensitivity | 3,595.23 | – | – | −6,354.68 | – |
| Pneumonitis/pneumonia | 7,716.51 | 5,753.29 | 8,207.10 | 5,997.89 | 10,221.94 |
| Polyneuropathy | 3,050.80 | 6,388.98 | −340.71 | 5,150.75 | 12,031.50 |
| Proteinuria | 6,974.85 | 4,195.63 | 3,538.41 | 7,872.77 | 15,813.17 |
| Pruritus/erythema/dermatitis | 8,849.81 | 3,167.64 | 4,370.41 | 3,178.01 | 10,620.25 |

*(Continued)*

**Table 3.** (Continued)

| Adverse event, $ | Breast (n = 117,322) | Lung (n = 55,834) | Colorectal (n = 45,051) | Skin (n = 42,804) | Non-Hodgkin lymphoma (n = 26,610) |
|---|---|---|---|---|---|
| Psychiatric disorders | 9,152.85 | 7,845.05 | 2,988.72 | 6,262.40 | 9,130.57 |
| Retinal/corneal/sclera problems | 7,352.14 | 7,294.46 | −4,091.36 | 8,117.46 | 2,502.73 |
| Sepsis | 10,804.36 | 7,742.44 | 7,114.58 | 9,818.53 | 9,326.05 |
| Stomatitis | 10,272.86 | 19,490.92 | −2,504.60 | −540.01 | 17,778.12 |
| Thrombocytopenia | 5,459.66 | 5,849.81 | 4,362.03 | 7,658.69 | 11,675.61 |
| Thromboembolic event[a] | 10,912.36 | 6,965.93 | 4,876.70 | 6,787.82 | 11,853.90 |
| Upper respiratory infections | 10,405.34 | 8,386.01 | 5,023.67 | 6,138.23 | 11,370.00 |
| Vomiting | 4,912.39 | 2,439.63 | 1,322.40 | 5,293.66 | 6,774.34 |
| Weight gain | 8,071.26 | 9,288.42 | 6,418.35 | 8,568.07 | 4,209.63 |
| Weight loss | 7,339.50 | 6,129.05 | 4,742.37 | 6,693.16 | 3,788.98 |

AEs, adverse events; CNS, central nervous system; GI, gastrointestinal.

Cell shading is white for those with >1,500 matched treatment episodes, light gray for those with 1,000–1,500, light peach for those with 500 to <1,000, and light rose for those with <500.

En dash indicates insufficient sample size for robust statistical model.

[a]Includes pulmonary embolism, thrombotic/embolic stroke, and venous/arterial thromboembolism/embolism.

A fully comprehensive comparison of our results with the existing literature is not feasible because we generate several hundred distinct point estimates (for example, the incremental cost of heart failure in lung cancer versus the incremental cost of severe pain in multiple myeloma). In general, and as would be expected, there is variability in both population and design in the existing literature. For example, Engel-Nitz et al. (2020) estimated the costs of several common AEs in first-line non-small cell lung cancer (NSCLC) [19]. This population differs from ours both in the restriction to first-line treatment and in the restriction to NSCLC, the most common type of lung cancer. In addition, the authors stratified their estimates by regimen type (chemotherapy versus immunotherapy versus chemotherapy+immunotherapy) but did not generate estimates incremental to a comparison group that did not experience AEs; together these differences make comparison of findings infeasible. A second study (Lal et al. 2022) examined the costs of adverse events in patients with hepatocellular carcinoma, the most common type of liver cancer, in a pooled Medicare Advantage population [14]. Their design differs from ours in that our study considers all types of liver cancer, including but not limited to HCC, thus precluding direct comparison of estimates.

There are several limitations in this study. The validity of the results of propensity score matching and regression analysis are conditional on the assumption that, after matching and modeling, there remain no factors that correlate to both the outcome (healthcare costs) and the treatment variable (the occurrence of the AE of interest). If this assumption is violated, that produces a source of bias in these results. Another limitation is that claims data contain limited information on the underlying cause of a diagnosis; therefore, AEs could not be directly linked to a treatment or underlying disease. Additionally, claims data do not provide information on the severity of disease (cancer stage) or course of disease. Furthermore, the population that was analyzed here was a pooled population of patients with either commercial insurance or Medicare Advantage. As a result, the external validity of this analysis is limited to that population. Costs for patients with other types of insurance, or with no insurance, are likely to differ from the estimates presented here. Finally, there was a limited sample of severe AEs and a limited number of many of the specific tumor types selected for analysis; as a result, there is greater uncertainty in our estimates of the cost of AEs in these populations.

**Table 4. Mean incremental costs (in US dollars) per treatment episode—AEs of any severity in rarer tumor types (<6% of selected patients). Each n indicates the number of episodes.**

| Adverse event, $ | Head and neck (n=20,947) | Pancreatic (n=20,149) | Prostate (n=18,357) | Liver (n=15,070) | Multiple myeloma (n=13,976) | Bladder (n=10,058) | CLL (n=3,518) | Follicular lymphoma (n=2,870) |
|---|---|---|---|---|---|---|---|---|
| Acute kidney failure | 678.49 | 6,883.87 | 1,210.45 | 4,428.24 | 12,012.33 | 6,039.26 | 9,211.67 | 13,251.53 |
| Agranulocytosis (neutropenia/leukopenia) | 2,052.63 | 2,437.22 | 4,678.35 | 3,640.87 | 12,688.41 | 6,948.50 | 7,827.08 | 13,488.18 |
| Allergic reaction and application/administration site reaction | – | – | – | – | – | – | – | – |
| Anemia | −395.09 | −46.90 | 6,418.10 | 1,858.70 | 14,836.66 | −121.45 | 10,667.45 | 11,109.01 |
| Arrhythmia | 1,662.11 | −1,667.05 | 235.53 | 8,459.45 | 13,599.63 | −1,838.25 | 20,823.13 | 11,552.96 |
| CNS hemorrhage | −2,702.64 | 7,729.83 | −1,337.06 | −2,155.70 | 8,737.45 | 4,908.49 | 6,017.08 | −1,709.87 |
| Chest pain/angina | −75.61 | −2,186.78 | 81.32 | 4,494.16 | 13,171.00 | −647.41 | 8,498.40 | 3,625.22 |
| Colitis | 1,978.59 | −1,553.93 | 5,023.69 | 3,925.66 | 12,192.12 | 1,547.04 | 22,144.12 | 18,022.34 |
| Constipation | 2,492.98 | 2,713.88 | 13,158.46 | 6,677.76 | 20,098.56 | 3,111.88 | – | – |
| Cystitis/urinary tract infection | 2,218.37 | 726.30 | 3,137.97 | −5,895.79 | 15,162.37 | 1,868.52 | 10,887.99 | 5,199.02 |
| Dehydration | −2,228.98 | −6,338.62 | −2,120.52 | 2,845.45 | 11,201.98 | 251.59 | 1,139.02 | 3,205.72 |
| Diarrhea | 1,043.96 | 2,231.18 | −3,666.53 | −888.56 | 15,454.21 | 3,628.27 | – | – |
| Dizziness | −3,027.12 | 1,840.65 | −729.63 | 3,697.33 | 2,359.80 | 3,885.10 | 707.24 | 9,204.90 |
| Dyspnea | 6,300.00 | 1,548.80 | 3,860.89 | −158.99 | 7,871.06 | 5,002.49 | 14,907.25 | 10,750.11 |
| Electrolyte imbalance | 3,087.46 | −2,517.13 | 3,995.85 | 2,157.05 | 13,501.56 | 5,532.40 | 6,153.71 | 2,488.03 |
| Elevated liver transaminase levels | – | −9,577.26 | – | 4,881.07 | – | −11,537.68 | – | – |
| Encephalopathy | – | – | – | – | – | – | – | – |
| Extravasation | – | – | – | – | – | – | – | – |
| Fatigue/generalized weakness/asthenia | 1,470.46 | −2,296.83 | 978.87 | −3,857.79 | 13,241.22 | 2,114.03 | 3,562.73 | 8,568.11 |
| GI/anal fistula | – | – | – | – | – | – | – | – |
| GI bleeding | 11,625.43 | 16,987.34 | 16,437.49 | 24,431.48 | 23,881.17 | 2,500.15 | – | – |
| GI perforation | – | – | – | – | – | – | – | – |
| Generalized edema | – | 11,587.49 | – | −8,544.59 | 10,518.36 | – | – | – |
| Headache/migraine | 1,272.83 | −7,457.95 | −889.37 | 8,100.25 | 14,962.34 | 2,552.60 | 29,364.80 | 14,790.12 |
| Hearing loss | – | – | – | – | – | – | – | – |
| Heart failure | – | 6,908.40 | 6,076.97 | −4,730.66 | 15,347.91 | 9,917.53 | – | – |
| Hematuria | 5,828.59 | 1,275.69 | 2,488.99 | 3,860.10 | 9,233.85 | 918.41 | 10,320.10 | 4,499.17 |
| Hyperglycemia | 3,108.07 | 18,951.73 | 189.36 | 358.67 | 7,352.43 | 6,763.57 | 25,927.74 | – |
| Hyperlipidemia | 6,362.37 | 683.16 | −1,116.52 | 7,592.66 | 8,340.68 | −3,004.71 | 3,354.53 | −615.37 |
| Hypertension | – | – | – | – | – | – | – | – |
| Hypotension | – | – | – | – | – | – | – | – |
| Hypothyroidism/hyperthyroidism | 7,557.83 | 17,196.44 | −1,790.38 | 20,609.68 | – | 1,999.52 | – | – |
| Hypoxemia | 6,968.12 | −2,623.40 | 13,826.46 | 6,908.56 | 8,629.45 | 18,892.76 | 29,790.45 | – |
| Ketoacidosis | – | – | – | – | – | – | – | – |
| Lymphocytopenia | – | – | – | – | 15,826.08 | – | – | – |
| Nausea | −478.13 | 1,216.42 | −4,869.59 | 710.11 | 9,985.26 | 6,187.28 | 7,565.52 | 6,092.35 |
| Pain | 2,793.31 | −22,271.41 | 1,226.33 | 3,871.79 | 13,623.97 | −1,505.53 | 10,337.42 | 12,468.58 |
| Pancreatitis | 8,065.38 | −2,829.69 | – | 31,783.93 | – | – | – | – |
| Pancytopenia | 4,967.65 | 7,104.68 | −4,023.21 | −5,661.94 | 15,510.33 | 8,256.02 | 30,018.10 | 32,941.12 |

*(Continued)*

| Adverse event, $ | Head and neck (n=20,947) | Pancreatic (n=20,149) | Prostate (n=18,357) | Liver (n=15,070) | Multiple myeloma (n=13,976) | Bladder (n=10,058) | CLL (n=3,518) | Follicular lymphoma (n=2,870) |
|---|---|---|---|---|---|---|---|---|
| Photosensitivity | – | – | – | – | – | – | – | – |
| Pneumonitis/pneumonia | 2,520.85 | 1,730.15 | 1,045.37 | 4,579.83 | 9,480.81 | 6,649.61 | 9,594.75 | 14,818.13 |
| Polyneuropathy | −3,072.51 | 7,749.74 | 19,991.99 | −5,746.63 | 9,296.09 | 11,443.68 | – | – |
| Proteinuria | −215.76 | 2,429.79 | −1,343.95 | 629.81 | 13,556.10 | 2,933.77 | 19,191.10 | −6,547.13 |
| Pruritus/erythema/dermatitis | −314.22 | −1,635.45 | 3,236.88 | 1,807.42 | 14,473.58 | −90.81 | 2,410.03 | −831.99 |
| Psychiatric disorders | 17.88 | 4,448.07 | 2,051.09 | 1,193.81 | 14,206.25 | 4,373.72 | 7,866.47 | 1,889.69 |
| Retinal/corneal/sclera problems | 6,295.02 | −3,755.37 | 1,784.14 | 10,873.38 | 15,515.72 | 5,011.53 | −5,080.06 | −1,918.59 |
| Sepsis | 7,728.41 | 493.50 | 6,979.59 | 5,963.05 | 11,377.64 | −3,625.34 | 10,764.94 | 14,398.21 |
| Stomatitis | 2,263.37 | 1,646.20 | – | −8,126.02 | −952.72 | – | – | – |
| Thrombocytopenia | −401.60 | 4,091.59 | 2,168.72 | 6,345.95 | 15,270.12 | 4,552.65 | 15,001.07 | 8,307.35 |
| Thromboembolic event | 10,108.10 | −616.23 | 4,434.61 | 2,298.96 | 14,140.23 | 2,884.20 | 12,914.84 | 7,378.80 |
| Upper respiratory infections | 186.27 | −138.33 | 1,416.36 | 7,686.29 | 11,690.38 | 3,230.78 | 11,032.60 | 1,697.06 |
| Vomiting | 184.94 | −319.14 | 3,674.85 | 3,851.54 | 11,910.52 | 2,945.53 | 6,926.00 | 2,203.55 |
| Weight gain | – | – | 2,829.18 | – | – | – | – | – |
| Weight loss | 1,911.26 | 112.46 | 5,555.58 | −2,964.51 | 13,484.73 | 1,907.65 | 8,598.13 | −19,596.61 |

AEs, adverse events; CLL, chronic lymphocytic leukemia; CNS, central nervous system; GI, gastrointestinal.

Cell shading is white for those with >1,500 matched treatment episodes, light gray for those with 1,000–1,500, light peach for those with 500 to <1,000, and light rose for those with <500.

En dash indicates insufficient sample size for robust statistical model.

[a]Includes pulmonary embolism, thrombotic/embolic stroke, and venous/arterial thromboembolism/embolism.

# 5. Conclusions

This study applied the design of Wong et al. to an updated patient data set with more tumor types and AEs and improved upon its methodology by applying the more up-to-date and more accurate ICD-10 diagnosis schema. These results are not intended to directly inform decision-making by payers, providers, or government actors. Rather, by providing detailed estimates of the cost of common, cancer-related AEs, this study improves the ability of health economists to accurately characterize incremental cost differences between different regimens for the treatment of cancer. It is our hope that the results generated here will be utilized as inputs into comparative models (e.g., cost-effectiveness models, treatment impact models) by the larger health economics community.

# Supporting information

**S1 Fig. Correlation matrix for AEs of any severity in breast cancer cohort.** AE, adverse event; CNS, central nervous system; GI, gastrointestinal. [a]Arrhythmia, i.e., arrhythmia/tachycardia/atrial fibrillation. [b]Pain includes myalgia, back/musculoskeletal/laryngeal pain, pain in extremities, proctalgia, pain in limb, abdominal pain, and neuritic pain. [c]Psychiatric disorders include depression, anxiety, confusion, agitation, euphoria, disorientation, emotional ability, hallucinations, mood alteration, nervousness, agitation, and irritability. [d]Thromboembolic events include pulmonary embolism, thrombotic/embolic stroke, and venous/arterial thromboembolism/embolism.
(TIF)

**S2 Fig. Correlation matrix for severe AEs[a] in breast cancer cohort.** AE, adverse event. [a]Only severe AEs (AEs grade 3–4) with available data were evaluated. [b]Arrhythmia, i.e., arrhythmia/tachycardia/atrial fibrillation. [c]Pain includes myalgia, back/musculoskeletal/laryngeal pain, pain in extremities, proctalgia, pain in limb, abdominal pain, and neuritic pain.

[d]Thromboembolic events include pulmonary embolism, thrombotic/embolic stroke, and venous/arterial thromboembolism/embolism.
(TIF)

## Acknowledgments

The authors are grateful to Justin Nedzesky for early contributions he made to the design of this study. We are also grateful to Will Wong for his feedback on the study protocol and on early versions of this manuscript. Medical writing support provided by Lidija Garan, PhD, of Nucleus Global, an Inizio company.

## Author contributions

**Conceptualization:** Jesse Sussell.

**Data curation:** Achal Patel, Robert Schuldt.

**Formal analysis:** Achal Patel, Robert Schuldt.

**Project administration:** Achal Patel, Robert Schuldt, Jesse Sussell.

**Software:** Achal Patel, Robert Schuldt.

**Supervision:** Jesse Sussell.

**Validation:** Achal Patel, Robert Schuldt.

**Visualization:** Achal Patel, Robert Schuldt, Jesse Sussell.

**Writing – review & editing:** Achal Patel, Robert Schuldt, Jesse Sussell.

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
