## [Decision Letter · Decision Letter 0]

7 Jan 2025

Dear Dr. Patel,

We look forward to receiving your revised manuscript.

Kind regards,

Mickael Essouma, M. D.

Academic Editor

PLOS ONE

Journal Requirements:

“This study was funded by Genentech, Inc.”

3. Thank you for stating the following in the Competing Interests/Financial Disclosure * (delete as necessary) section:

“I have read the journal's policy and the authors of this manuscript have the following competing interests: Stock or stock options- Genentech; All support for the present manuscript (e.g., funding, provision of study materials, medical writing, article processing charges, etc.) - Genentech; current employer- Genentech”

We note that you received funding from a commercial source: Genentech

Reviewers' comments:

Reviewer's Responses to Questions

**Comments to the Author**

1. Is the manuscript technically sound, and do the data support the conclusions?

Reviewer #1: Partly

Reviewer #2: Yes

Reviewer #3: Yes

2. Has the statistical analysis been performed appropriately and rigorously?

Reviewer #1: I Don't Know

Reviewer #2: Yes

Reviewer #3: Yes

3. Have the authors made all data underlying the findings in their manuscript fully available?

Reviewer #1: No

Reviewer #2: Yes

Reviewer #3: Yes

4. Is the manuscript presented in an intelligible fashion and written in standard English?

Reviewer #1: Yes

Reviewer #2: Yes

Reviewer #3: Yes

Reviewer #1: Thank you for the opportunity to review this paper. I have the following comments.

1. First, more information on methodology should be provided. Just indicating to read another paper throughout is not sufficient.

2. How are the AEs identified ? Is it in any claims line? Or just in position 1 or 2?. If it is in any claim line, such as claim line 6, then the cost difference less likely due to this event and also maybe a chronic condition. The difference in costs then maybe negative due to the fact the control patient has some other more expensive condition occurring at the same time.

3. How are patients who died handled?

4. Were diagnostic claims included in selection of patients?

5. How were maintenance lines of treatment considered? Their rates of AEs and costs are difference than during active treatment regimens.

6. Are these paid costs? What year were they adjusted to? Are these cost per episode? Are the episodes in similar length? Should have calculated PPPMs or used a fixed time period of follow-up or even a fixed line of therapy( eg.first line).

7. There are patients who are censored and have unaccounted time period for AE occurrences and AE costs. Would need to conduct sensitivity analysis with patients with completed episodes.

8. It may be good to have a figure to explain baseline, episode start and episode end in active and control patient.

9.For Severe AE's which involve hospitalization, what line did the AE have to be in? If the person came in for a car accident and only had the AE in line 6, then the impact on costs would be different. It would be better to have the AE in the first or second position in the claims record to say the AE is the main reason for the hospitalization.

10. Not sure truncation is a good way to set the time period of the control, especially when patients could be on different treatments (drugs) of different lengths and there is no matching on the exact regimen, as I read the matching section. I would think this is a major limitation, especially since we seem to be front loading the regimen period and some adverse events take place later in the line or even after the end of the line. Sometimes, AEs can occur up to 180 days after the end of a regimen. These are limitations to be added to the discussion.

11. What is regimen line of therapy number in line 195? Since progression is not captured in claims, any regimen change is a line of therapy change. Regimen was not matched on, so episodes can be very different types of drugs and lengths and just truncating time period does not overcome this limitation.

12. Assume a GLM was conducted for the multivariable analysis. It is not stated.

13.These negative values are problematic. It may make sense from a model perspective, but not from a clinical and economic perspective.

14. The conclusion of negative values does not quite make sense. If the control have other conditions which were more expensive, that maybe driving the negative values. There is no matching on comorbidities conducted.

15. Line 201 - 47? Wouldn’t this be 53, since there were a total of 54? Or were some correlated ones taken out?

Reviewer #2: The article is well-written, and the paper's statistical analysis is well done. However, the authors use some statistical models/methods (mainly bias reduction techniques). As the authors partially state in lines 326-329, all these statistical methods are based on underlying assumptions. The authors should first express these assumptions and then conveniently validate them based on the available data. Model validation (i.e., evaluating whether a chosen statistical model is appropriate) is as essential as model fitting in a good statistical analysis.

Reviewer #3: The article, "New Estimates of the Costs of Adverse Events in Patients with Cancer", is a commendable contribution and a strong candidate for publication in this journal. It aligns well with the journal's standards for original research by employing a detailed methodology, rigorous statistical analysis, and a comprehensive discussion of the findings. The study addresses a significant gap in the literature by evaluating the incremental costs of adverse events (AEs) in cancer treatment across multiple tumour types, using the updated ICD-10 schema and recent data. The authors' sophisticated approach to bias control and their detailed stratification enhance the study’s credibility and relevance, making it a valuable resource for the field.

Abstract

The abstract effectively summarises the study’s objectives, methodology, and key findings, and successfully conveys its significance in understanding the economic burden of AEs in cancer care. However, it lacks balance, as it does not mention the study’s limitations, which would provide a more rounded perspective. Furthermore, the abstract does not explore the broader implications of the findings, such as their potential applications in healthcare policy or economic modelling. A brief acknowledgement of the study's limitations and practical applications would significantly enhance its comprehensiveness and appeal to readers.

Introduction

The introduction establishes a strong foundation by justifying the need to quantify AE costs and contextualising the research within the broader field of healthcare economics. It effectively underscores the relevance of these costs for guiding payer decisions and enhancing economic models. However, it misses the opportunity to clearly articulate the unique contributions of this study compared to prior research, particularly the work of Wong et al. Explicitly identifying the research gaps being addressed—such as the broader range of tumour types, updated datasets, and advanced methodologies—would further strengthen this section.

Materials and Methods

This section demonstrates strong scientific rigour, employing techniques such as propensity score matching and exact tumour-type matching to minimise bias. The use of updated ICD-10 codes and recent datasets ensures the relevance and applicability of the findings. However, the exclusion of treatment-related costs, such as surgery and pharmaceutical expenses, is insufficiently justified, potentially limiting the scope of the analysis. Additionally, the assumptions underlying the matching process and the inherent biases of claims data are not critically evaluated. A more thorough discussion of these methodological decisions and their implications would enhance the robustness of this section.

Results

The results section is well-organised and provides detailed insights into AE-related costs across different cancer types and severity levels. The stratification by tumour type and severity makes the findings highly applicable to diverse clinical and economic contexts. However, the meaning of negative cost estimates is not adequately clarified, which may confuse readers. Furthermore, the broader significance of the findings for healthcare planning and policy remains underexplored. Adding a clear interpretation of negative cost estimates and summarising the key findings with practical implications would improve the impact of this section.

Discussion

The discussion effectively links the findings to the study’s objectives and prior literature, highlighting the advancements in methodology and the importance of AE cost estimates. It also acknowledges important limitations, such as reliance on claims data and small sample sizes for rare cancers. However, the section falls short in exploring the practical applications of the findings, such as guiding cost management strategies or informing healthcare policies. Additionally, while variability in AE costs for rare cancers is noted, it is not explored in sufficient depth. Addressing these gaps would provide a more comprehensive understanding of the study’s implications.

Conclusion

The conclusion succinctly underscores the study’s contributions to health economic modelling and the value of detailed AE cost estimates for decision-making. However, it does not provide actionable recommendations for policymakers, healthcare providers, or researchers, nor does it fully address the broader significance of the findings in improving resource allocation or optimising treatment planning. Including specific recommendations for clinical and policy applications, as well as suggestions for future research directions, would make the conclusion more impactful and forward-looking.

References

The references section is thorough and draws upon relevant and recent literature to substantiate the study’s findings. Broadening the range of cited works to include more diverse research approaches would enrich the study’s contextual foundation and demonstrate a more comprehensive engagement with the field.

Overall

The article is well-structured, scientifically robust, and addresses a critical gap in health economic modelling for cancer care. However, its conclusions in each section could better emphasise actionable outcomes, reflect on methodological limitations, and explore broader implications for healthcare policy and practice. By incorporating more explicit recommendations and transparently addressing the study’s limitations, the article could significantly enhance its relevance and impact within the field.

**Do you want your identity to be public for this peer review?** For information about this choice, including consent withdrawal, please see our Privacy Policy

Reviewer #1: No

Reviewer #2: No

Reviewer #3: No

---

## [Author Response · Author response to Decision Letter 1]

15 Apr 2025

Mickael Essouma, MD

Academic Editor, PLOS One

April 2025

Dear Dr. Essouma:

On behalf of the authors, we are pleased to resubmit our revised manuscript, “New estimates of the costs of adverse events in patients with cancer” to PLOS One.

We appreciate the constructive comments from the reviewers and have addressed each one in the following detailed responses along with tracked changes marked in the revised manuscript.

As requested, following please find the amended Role of the Funder and Competing Interests statements:

Role of the Funder

This study was funded by Genentech, Inc. The sponsor participated in the design, analysis, and interpretation of the study.

Competing Interests

All authors have read the Journal's policy and the authors of this manuscript have the following competing interests: Employment and stock or stock options in Genentech, Inc. All support for the present manuscript (e.g., funding, provision of study materials, medical writing, article processing charges, etc.) was provided by Genentech, Inc. This does not alter our adherence to PLOS ONE policies on sharing data and materials.

Thank you for your consideration, we look forward to hearing from you.

Sincerely,

Achal Patel, PhD, on behalf of the authors

patel.achal@gene.com

Journal Requirements

https://journaIs.plos.org/ plosone/s/file?id=wjVg/PLOSOne_formatting_sample_main_body.pdf and https://journaIs.plos.org/plosone/s/file?id=ba62/PLOSOne_formatting_sample_title_authors_affiIiations.pdf

Authors’ response: Thank you, we have confirmed and made minor adjustments where needed to align with the published style.

2. Thank you for stating the following financial disclosure: "This study was funded by Genentech, Inc." Please state what role the funders took in the study. If the funders had no role, please state: "The funders had no role in study design, data collection and analysis, decision to publish, or preparation of the manuscript." If this statement is not correct you must amend it as needed. Please include this amended Role of Funder statement in your cover letter; we will change the online submission form on your behalf.

Authors’ response: We have amended the Role of the Funder statement and included it in the cover letter.

3. Thank you for stating the following in the Competing Interests/Financial Disclosure* (delete as necessary) section: "I have read the journal's policy and the authors of this manuscript have the following competing interests: Stock or stock options- Genentech; All support for the present manuscript (e.g., funding, provision of study materials, medical writing, article processing charges, etc.) - Genentech; current employer- Genentech"

We note that you received funding from a commercial source: Genentech

Within this Competing Interests Statement, please confirm that this does not alter your adherence to all PLOS ONE policies on sharing data and materials by including the following statement: "This does not alter our adherence to PLOS ONE policies on sharing data and materials." (as detailed online in our guide for authors http://journals.plos.org/plosone/s/competing-interests). If there are restrictions on sharing of data and/or materials, please state these. Please note that we cannot proceed with consideration of your article until this information has been declared.

Authors’ response: We have amended the Competing Interests statement and included it in the cover letter.

Reviewer #1

Thank you for the opportunity to review this paper. I have the following comments.

1. First, more information on methodology should be provided. Just indicating to read another paper throughout is not sufficient.

Authors’ response: Thank you for this feedback. We have revised our methods section to provide more complete information on our analytical approach. We do make several references throughout the paper to the Wong study, which we believe is appropriate, as our study applies a similar design to a new dataset. On review of the paper, we have identified one instance where we suggested the reader refer to the Wong paper for more methodological detail (section 2.6, methods). We have changed the presentation to clarify that while the methods are similar, (A) we summarize those methods in detail here, and (B) there are several notable differences, which are described.

2. How are the AEs identified? Is it in any claims line? Or just in position 1 or 27. If it is in any claim line, such as claim line 6, then the cost difference less likely due to this event and also maybe a chronic condition. The difference in costs then maybe negative due to the fact the control patient has some other more expensive condition occurring at the same time.

Authors’ response: AEs are identified based on ICD-10-CM codes recorded on a health insurance claim in any medical setting (e.g., inpatient stay, emergency room visits, outpatient visits) and any diagnosis position (In P+ indicated by diag1-diag12). We have added the diagnosis position randomness as a potential explanation for the observed negative point estimates.

3. How are patients who died handled?

Authors’ response: Patient death is not well measured in claims data. In the cohort selection process, the end of all treatment episodes is defined as the occurrence of either initiation of a new antineoplastic regimen, administrative censoring (end of continuous enrollment), or end of data availability. Death is an event which triggers the end of enrollment; thus in the context of this study death is best understood (correctly) as an event which precipitates the end of cost measurement for any given treatment episode. We have updated the manuscript to reflect this.

4. Were diagnostic claims included in selection of patients?

Authors’ response: Yes, as we describe in the methods section, the treatment group cohort is essentially defined by the presence of diagnosis codes for adverse events of interest. Additionally (as described), because the control group is defined by the absence of diagnosis codes for AEs of interest, and because we required exact matching on tumor type, diagnosis codes are a critical component of control group selection.

5. How were maintenance lines of treatment considered? Their rates of AEs and costs are difference than during active treatment regimens.

Authors’ response: Thank you for this comment. We believe that the existing design controls for the potential bias you describe. This is because our design requires matching on regimen (see response to comment #10, below). Thus any differences that arise in adverse event costs due to differences in regimen (including maintenance vs. active treatment) are theoretically controlled for.

6. Are these paid costs? What year were they adjusted to? Are these cost per episode? Are the episodes in similar length? Should have calculated PPPMs or used a fixed time period of follow-up or even a fixed line of therapy (e.g., first line).

Authors’ response: The estimates presented in this paper describe the average cost for a given instance of an adverse event of interest. The quantity being averaged is the total incremental difference in "allowed" claims, not paid claims. So it is not an average of the amount the insurers paid out, but rather an average of the contractually agreed upon amounts that insurers will pay. We believe when the reviewer refers to episodes, the reviewer is describing episodes of adverse events, rather than episodes of treatment (described in detail in the paper). If this is correct, then it is not known whether adverse event episodes are similar in length, because this is not a quantity that is measured in the data (unlike, for example, hospital stays or duration on antineoplastic therapy).

We believe this is the best way to present the results. PPPMs can readily be calculated by any payer interested in that quantity. Selection of a specific duration for a fixed time period analysis would jeopardize external validity either by omitting relevant costs (if the time period is too short) or by running afoul of the treatment episode construction (if the time period is too long).

We agree that stratification of results by line of therapy could be interesting and would certainly be a valid research question. Although we controlled for line of therapy in our propensity matching analysis, we felt that the existing stratification factors (tumor type, all AEs vs severe AEs) were more important to focus on, and that adding an additional level of stratification would make the overall volume of results presented unwieldy. This would be an interesting question for a future study.

7. There are patients who are censored and have unaccounted time period for AE occurrences and AE costs. Would need to conduct sensitivity analysis with patients with completed episodes.

Authors’ response: The objective of the present analysis is to estimate the incremental cost of the occurrence of a given adverse event in patients with cancer, not to estimate the total cost of cancer care. While the latter construct clearly may differ between censored and uncensored patients, there are no a priori reasons to think that the former does. Additionally, many treatments for metastatic cancer are "treat to progression;" because neither metastatic (versus early) stage disease status nor the occurrence of progression are well measured in claims data, it is not possible to define "completed" treatment regimens in claims data in a consistent way spanning the thousands of different tumor / regimen combinations assessed in this study.

8. It may be good to have a figure to explain baseline, episode start and episode end in active and control patient.

Authors’ response: This information is provided graphically in Figure 2 with detailed definitions provided in the footnote. We would be happy to revise the figure if the reviewer will clarify what aspects are unclear.

9. For Severe AE's which involve hospitalization, what line did the AE have to be in? If the person came in for a car accident and only had the AE in line 6, then the impact on costs would be different. It would be better to have the AE in the first or second position in the claims record to say the AE is the main reason for the hospitalization.

Authors’ response: Thank you for this comment. We confirm that for the analysis of Severe AEs, the AE of interest must be the reason for admission, operationalized as position one or two. We have clarified this in the manuscript.

10. Not sure truncation is a good way to set the time period of the control, especially when patients could be on different treatments (drugs) of different lengths and there is no matching on the exact regimen, as I read the matching section. I would think this is a major limitation, especially since we seem to be front loading the regimen period and some adverse events take place later in the line or even after the end of the line. Sometimes, AEs can occur up to 180 days after the end of a regimen. These are limitations to be added to the discussion.

Authors’ response: Thank you, we see that we inadvertently omitted the word "regimen" out of the list of factors included in the PSM. We have corrected this omission.

With matching on regimen, we believe the concern about truncation is removed. In particular, it is desirable to have the duration of the control episode set to be equal to the duration of the matched treatment episode. In general, accrued costs increase as a function of the length of the treatment episode. If control group episodes are longer than treatment group episodes, they will include costs unrelated to AEs of interest, and our estimates will be biased negative; if control group episodes are shorter than treatment group episodes, the reverse will be true.

11. What is regimen line of therapy number in line 195? Since progression is not captured in claims, any regimen change is a line of therapy change. Regimen was not matched on, so episodes can be very different types of drugs and lengths and just truncating time period does not overcome this limitation.

Authors’ response: This is related to our response to comment 10 above, regarding whether the regimen was controlled for. "Regimen line of therapy" was the variable name in our analytic dataset for the construct 'regimen' - specifically "regimen_lot". This variable consists of a categorical variable which is constructed by alpha sorting the agents within a given treatment episode, such that, for example, (pertuzumab, trastuzumab) will be correctly matched to (trastuzumab, pertuzumab).

The manuscript had erroneously included the text "line of therapy" as part of the descriptor for this variable, which we have corrected. So to clarify, both “regimen” and “line of therapy” were matched on. We apologize for the confusion around this.

12. Assume a GLM was conducted for the multivariable analysis. It is not stated.

Authors’ response: Thank you for pointing out this omission. In fact, we used OLS rather than GLM. In early analysis we considered GLM, but encountered significant model convergence issues, particularly when attempting to generate point estimates for tumor types with smaller sample sizes. We have revised the manuscript to make this clear.

13. These negative values are problematic. It may make sense from a model perspective, but not from a clinical and economic perspective.

Authors’ response: We agree with the reviewer’s point and have added further comments in the Discussion indicating that the negative values should be interpreted with caution. We have also added significant additional text providing potential explanations for why these results are observed.

14. The conclusion of negative values does not quite make sense. If the control have other conditions which were more expensive, that maybe driving the negative values. There is no matching on comorbidities conducted.

Authors’ response: Thank you for this valuable suggestion. We agree that comorbidities are a potential source of confounding. We have added the Charlson Comorbidity Index as a control variable to the outcomes model and revised the entire presentation of results to reflect that.

Even after controlling for co-morbidities in our revised models, there are rare instances where there is a negative incremental cost associated with experiencing an AE (e.g., across all cancers, photosensitivity is associated with $-488 incremental cost relative to the control group). One possible explanation is differential co-occurrence of other AEs (including AEs not directly measured in this analysis) for the AE vs. control group, where more expensive AEs (measured or unmeasured) are more common among the control group. In their analysis, Wong et al. noted this as a potential explanation for rare negative AE incremental cost values.

While we attempt to reduce the impact of other co-occurring AEs on the incremental cost of any given AE via our outcomes model, the impact of potential unmeasured co-occurring AEs is a limitation of the study that we acknowledge. Another potential explanation is that negative incremental cost values are more likely with reduced sample sizes, due to the higher sample variance involved in estimation. Because of this high variance, the sample means (and the incremental cost difference computed based on the sample means) may differ from the true, population mean differences. We’ve also described this potential limitation in detail in the conclusions section of the manuscript.

15. Line 201 - 47? Wouldn't this be 53, since there were a total of 54? Or were some correlated ones taken out?

Authors’ response: Thank you, yes, we have

---

## [Decision Letter · Decision Letter 1]

4 Jun 2025

New estimates of the costs of adverse events in patients with cancer

PLOS ONE

Dear Dr. Patel,

Thank you for submitting your manuscript to PLOS ONE. After careful consideration, we feel that it has merit but does not fully meet PLOS ONE’s publication criteria as it currently stands. Therefore, we invite you to submit a revised version of the manuscript that addresses the points raised during the review process.

We look forward to receiving your revised manuscript.

Kind regards,

Monia Marchetti

Academic Editor

PLOS ONE

Reviewers' comments:

Reviewer's Responses to Questions

**Comments to the Author**

Reviewer #1: (No Response)

Reviewer #2: All comments have been addressed

2. Is the manuscript technically sound, and do the data support the conclusions?

Reviewer #1: Yes

Reviewer #2: Yes

3. Has the statistical analysis been performed appropriately and rigorously?

Reviewer #1: Yes

Reviewer #2: Yes

4. Have the authors made all data underlying the findings in their manuscript fully available?

Reviewer #1: Yes

Reviewer #2: Yes

5. Is the manuscript presented in an intelligible fashion and written in standard English?

Reviewer #1: Yes

Reviewer #2: Yes

Reviewer #1: Thank you for the opportunity to re-review the manuscript. I believe most of my comments have been addressed. Two issues remain:

1. Diagnostic claims are claims associated with determining the diagnosis and include the initial imaging, biopsy, pathology tests and associated codes. Were these included in determine patient selection?

2. There is a summary of a paper by Lal et al in the discussion, which is not summarized appropriately. The study population is not just Medicare Advantage; it also includes commercial patients.

Reviewer #2: (No Response)

**Do you want your identity to be public for this peer review?** For information about this choice, including consent withdrawal, please see our Privacy Policy

Reviewer #1: No

Reviewer #2: No

---

## [Author Response · Author response to Decision Letter 2]

19 Jul 2025

Reviewer #1: Thank you for the opportunity to re-review the manuscript. I believe most of my comments have been addressed. Two issues remain:

1. Diagnostic claims are claims associated with determining the diagnosis and include the initial imaging, biopsy, pathology tests and associated codes. Were these included in determine patient selection?

Response:

Thank you for this question. To clarify, patient selection was determined through the ICD10 diagnosis classification associated with the medical and pharmacy claims themselves. So for example, the patients in our lung cancer cohort had pharmacy/medical claims with the C34.x ICD10 codes listed in the diagnosis position.

It is true that many of these patients had diagnostic procedures associated with their conditions, and that these procedures were used to determine the actual diagnosis of lung cancer. For example, a patient with lung cancer may have had an MRI during the course of diagnosis. But a patient with COPD might also have an MRI. If this had been an analysis of Electronic Medical Record (EMR) data, this information could be utilized to inform patient selection / classification. But the nature of claims data is that at the time the MRI (or biopsy, or panel blood test) occurs, the actual diagnosis is not yet known: Someone who has a chest MRI may have lung cancer, or COPD, or any of several other conditions.

To summarize, we did not use the presence of specific diagnostic tests to determine patient cohort classification, because in claims data (at least at the beginning of the treatment journey) the information associated with those claims does not typically convey information about the patient's actual diagnosis.

2. There is a summary of a paper by Lal et al in the discussion, which is not summarized appropriately. The study population is not just Medicare Advantage; it also includes commercial patients.

Response:

Thank you for bringing this error to our attention. We have revised the manuscript text as follows: A second study examined the costs of adverse events in patients with hepatocellular carcinoma, the most common type of liver cancer, in a pooled Medicare Advantage population. Their design differs from ours in that our study considers all types of liver cancer, including but not limited to HCC, thus precluding direct comparison of estimates.

---

## [Editor Report · Decision Letter 2]

4 Sep 2025

New estimates of the costs of adverse events in patients with cancer

PONE-D-24-53463R2

Dear Dr. Patel,

We’re pleased to inform you that your manuscript has been judged scientifically suitable for publication and will be formally accepted for publication once it meets all outstanding technical requirements.

Kind regards,

Monia Marchetti

Academic Editor

PLOS ONE
---

## [Editor Report · Acceptance letter]

PONE-D-24-53463R2

PLOS ONE

Dear Dr. Patel,

I'm pleased to inform you that your manuscript has been deemed suitable for publication in PLOS ONE. Congratulations! Your manuscript is now being handed over to our production team.

Kind regards,

on behalf of

Dr. Monia Marchetti

Academic Editor

PLOS ONE